# Interplay between mobility, multi-seeding and lockdowns shapes COVID-19 local impact

**Mattia Mazzoli**[1,4]*, **Emanuele Pepe**[2], **David Mateo**[3], **Ciro Cattuto**[2], **Laetitia Gauvin**[2], **Paolo Bajardi**[2], **Michele Tizzoni**[2], **Alberto Hernando**[3], **Sandro Meloni**[2], **José J. Ramasco**[1]*

**1** Instituto de Física Interdisciplinar y Sistemas Complejos IFISC (CSIC-UIB), Palma de Mallorca, Spain, **2** ISI Foundation, Turin, Italy, **3** Kido Dynamics SA, Lausanne, Switzerland, **4** INSERM, Sorbonne Université, Institut Pierre Louis d'Epidémiologie et de Santé Publique, IPLESP, Paris, France

* mattia.mazzoli@inserm.fr (MM); jramasco@ifisc.uib-csic.es (JJR)

**Data Availability Statement:** The information on mobility in Spain is extracted from mobile phone records based on antenna service (detailed description included in the Methods section). The

## Abstract

Assessing the impact of mobility on epidemic spreading is of crucial importance for understanding the effect of policies like mass quarantines and selective re-openings. While many factors affect disease incidence at a local level, making it more or less homogeneous with respect to other areas, the importance of multi-seeding has often been overlooked. Multi-seeding occurs when several independent (non-clustered) infected individuals arrive at a susceptible population. This can lead to independent outbreaks that spark from distinct areas of the local contact (social) network. Such mechanism has the potential to boost incidence, making control efforts and contact tracing less effective. Here, through a modeling approach we show that the effect produced by the number of initial infections is non-linear on the incidence peak and peak time. When case importations are carried by mobility from an already infected area, this effect is further enhanced by the local demography and underlying mixing patterns: the impact of every seed is larger in smaller populations. Finally, both in the model simulations and the analysis, we show that a multi-seeding effect combined with mobility restrictions can explain the observed spatial heterogeneities in the first wave of COVID-19 incidence and mortality in five European countries. Our results allow us for identifying what we have called epidemic epicenter: an area that shapes incidence and mortality peaks in the entire country. The present work further clarifies the nonlinear effects that mobility can have on the evolution of an epidemic and highlight their relevance for epidemic control.

## Author summary

Human mobility controls the spreading of infectious diseases worldwide. Pathogens use infected individuals as vehicles to travel from one city to another, between countries and even across continents. We know that the arrival of the first case or seed at a population is connected to the probability of traveling there from the area of disease emergence. The question that we address here is not when the first cases arrive or the local outbreak starts, but whether the continuous arrival of more infected individuals can have an impact on

data is proprietary. Interested researchers will be able to obtain access to the aggregated mobility flows used in this work in the same way as the authors did upon request to Kido Dynamics SA (www.kidodynamics.com) through Ignacio Barrios (ibarrios@kidodynamics.com). Mobility data in Italy is publicly available through Ref. [38]. Mobility data in France, Germany, Spain, and the United Kingdom, is available from Cuebiq through their Data for Good program (https://www.cuebiq.com/about/data-for-good/). The GDPR-compliant data was accessed under license for this work. Interested researchers will be able to obtain access to the aggregated mobility flows used in this work in the same way as the authors did upon request to Cuebiq (dataforgood@cuebiq.com). The data on incidence and mortality are available at: England (https://coronavirus.data.gov.uk/#category= utlas&map=rate), France (https://www. europeandataportal.eu/data/datasets/chiffres-cles-concernant-lepidemie-de-covid19-en-france? locale=en, https://geodes.santepubliquefrance.fr/ #c=home), Germany (https://github.com/jgehrcke/ covid-19-germany-gae), Italy (https://github.com/ pcm-dpc/COVID-19) and Spain (https://github. com/montera34/escovid19data). The population counts per NUTS areas come from the national statistics offices: England (www.ons.gov.uk/), France (www.insee.fr), Germany (www.destatis. de), Italy (dati.istat.it) and Spain (www.ine.es).

**Funding:** M.M.'s salary was funded by the Conselleria d'Innovació, Recerca i Turisme of the Government of the Balearic Islands and the European Social Fund with grant code FPI/2090/ 2018. M.M., S.M. and J.J.R. also acknowledge funding from the project Distancia-COVID (CSIC-COVID-19-039) of CSIC integrated in the platform PTI Salud Global and funded by a contribution of AENA, also from the Spanish Ministry of Science, Innovation and Universities, the AEI and FEDER (EU) under the grant PACSS (RTI2018-093732-B-C22) and the Maria de Maeztu program for Units of Excellence in R&D (MDM-2017-0711). M.M. acknowledges financial support of the Sorbonne Université Emergence project RISKFLOW. E.P., L. G., C.C. and M.T. gratefully acknowledge the support of the Lagrange Project of the ISI Foundation funded by CRT Foundation. P.B. acknowledges support from Intesa Sanpaolo Innovation Center. The funders had no role in study design, data collection and analysis, decision to publish, or preparation of the manuscript.

**Competing interests:** The authors have declared that no competing interests exist.

the development of the local outbreak. We show with standard epidemic spreading models that indeed there is a relation between the number of seeds arriving at a location over the resident population, the height of the local incidence peaks and the total population finally affected. It is a non-linear relation, and it depends on the details of the social contact network in the destination area. After this theoretical work and thanks to mobility data from different European countries of Europe, we find that there are solid signs of multiseeding effects similar to those observed in the models in the propagation of the first COVID-19 wave in the continent. We take advantage of this to propose a method to understand and reconstruct the spatial spreading patterns of the main outbreak-producing events in every country. From a public health point of view, surveillance on the importation of cases in a region is fundamental to anticipate the severity of local outbreaks and minimize their consequences.

## Introduction

The COVID-19 epidemic reached the WHO status of pandemic on March 11, 2020 [1] and currently involves most of the countries of the world [2]. In Europe, SARS-CoV-2 severely hit Italy, the first country to report local transmission in mid/late-February [3], and by mid March many other countries such as France [4], Germany [5], Spain [6] and the UK [7] declared local outbreaks as well. As an illustrative example, Madrid was the first city and region with an important number of local contagions in Spain. Measures to prevent the propagation of the disease were implemented, first locally, as closing schools and universities, public buildings, etc, and, secondly, at national level with a population confinement at home (lockdown) [8] implemented on March 14, 2020, which with different degrees of severity lasted until June 21, 2020. Madrid, besides being the administrative capital, is also the main communication hub of the country, attracting workers from neighboring areas and students from all over Europe. The initial phases of the lockdown included a reduction of the frequencies and capacities of public transportation lines, with the most strict mitigation measures taken between March 29 and April 23. The situation in the other European countries evolved more or less in parallel with some delays or advances depending on the local propagation patterns.

The role of human mobility in shaping epidemic dynamics has been extensively considered [9–20], even recently for COVID-19 [21–32]. Pathogens use infected humans as vehicles and, consequently, disease propagation patterns strongly resemble transportation networks [9, 16, 33]. If a certain geographical region suffers a local outbreak, it is natural to expect that closely connected areas will start to import infected individuals. Closures, travel restrictions and lockdowns may help to delay the propagation, even though their efficacy strongly depends on disease etiology and epidemiological features [13, 15, 17, 23, 34–42].

Much less attention has been paid, however, to the impact of multi-seeding on the evolution of an epidemic. The time of arrival of the first infected individual (seed) and the most likely propagation pathways were analyzed in [33, 43–45]. The probability of traveling in models is proportional to the trip outflows from a given area. More trips arriving from the area where the initial outbreak developed imply a larger probability of receiving a first seed. Therefore, mobility from the original focus has been so far essential to calculate the risk of a certain population to suffer an outbreak [21, 46]. From a more theoretical viewpoint, Refs. [47, 48] show that the initial presence of more seeds may affect the critical infectivity needed to start the local epidemic. And, recently, it has been postulated that the stronger connection to the original focus of infection can favor a faster initial growth of the epidemic curves [21].

In this work, we focus on the full extent of the epidemic curve, and show that the arrival of more seeds does not only advance the emergence of local outbreaks but makes them more severe. We find a connection between the number of arriving seeds, the height of the local incidence peaks and the final population affected by the outbreak (size). Our conclusions rely upon two pillars: computational models and empirical findings. On the theoretical side, we analyze a model with different social contact networks, covering a full continuous range between well-mixed populations and a grid, in which space plays a leading role. The model results are analyzed first in a single population, then in two and finally in a full metapopulation framework with populations and mobility informed with real data from Spain. In a single population, we observe a nonlinear relation between the number of seeds, the height of the incidence peak and the time to the peak. We are able to collapse the epidemic curves with simple scaling arguments depending only on the dimensionality of the contact network to obtain a single common curve. Secondly, we simulate the spreading in two populations allowing mobility between the two areas. Importing more seeds strongly favors the development of severe outbreaks in the second population. In a metapopulation model with heterogeneity, this nonlinear relation with the number of seeds extends to all the studied epidemic variables. A key point to note here is that the relevant variable is not the crude number of arriving seeds, but rather their ratio over the local population. Attaining the same epidemic impacts requires the arrival of more seeds at a large metropolis than at a small isolated town. Finally, we confirm these theoretical results with an empirical analysis based on combining detailed anonymized and privacy enhanced human mobility data with epidemiological reports on COVID-19 in five European countries. We study the connection between inter-city mobility flows and incidence curves to find a clear relation between mobility and COVID-19 incidence peaks across Europe. Furthermore, a correlation analysis driven by mobility helps to describe the spatio-temporal progress of the disease in the first epidemic wave.

## Materials and methods

We introduce the different ingredients needed to build the models, the simulations of lockdowns, the data (geographical extensions considered, mobility and epidemic records) and, finally, the metrics employed to search for the areas causing country-scale outbreaks in the first wave of the COVID-19 pandemic.

### Basic structure of the models

With the aim of gaining insights into the role of multi-seeding in the spreading of a disease, we build a family of models within a metapopulation framework. Respect to the disease, the individuals can be susceptible ($S$), exposed ($E$), infected prodromic ($I_p$) (they have not developed symptoms yet but they are infectious), infected symptomatic ($I_s$), infected asymptomatic ($I_a$) and recovered ($R$). The model is a simplified version of the one introduced in Ref. [49] for COVID-19, but changing the parameters it can be valid as well for other airborne diseases such as the influenza. Inside each population, we consider four contacts per individual. We selected this number to keep coherence across networks. Since each population is going to represent a full province or geographical region, the space can play a role in how the social contacts occur. We wish to explore contact networks laying between two extreme cases: a 2D regular lattice (GRID), for which distance and space are essential, and a well mixed (WM) population, in which the four contacts are randomly selected every time step without any spatial structure. In order to interpolate between these two extreme configurations, we have randomly rewired couples of links in the GRID with a given probability $p$ generating the REW networks.

This introduces long-range links, reducing the distance between individuals and inducing a small world behavior for $p$ values of the order of $10^{-2}$ [50].

The average time an agent spends in each disease compartment is: $\tau_E = 3.7$ days for exposed, $\tau_p = 1.5$ days for prodromic, and $\tau_i = 2.3$ days for infected symptomatic or asymptomatic to pass to the recovery compartment. Prodromic individuals can become infected asymptomatic with a probability $p_a = 0.36$ and, otherwise, with probability $1 - p_a$, they become symptomatic. As in [49], the probability rate of infection of a susceptible agent in contact with an infected symptomatic individual is $\beta = 0.19$ days$^{-1}$ for WM and $\beta = 1.19$ days$^{-1}$ for the GRID contact networks (i.e., 2D regular lattices) in such a way the final epidemic size gets over 0.8 and all the realizations have an outbreak. We have tested other (lower) values of $\beta$ with similar results (see Figs Q-W in S1 Text). The infectivity gets reduced by a factor 0.55 if the infectious individual is asymptomatic or prodromic.

Our purpose is to have a stylized model, not a realistic one. Even so, as a structure for the populations is needed, we take the Spanish provinces as the basis for the meta-populations, where each node hosts a number of individuals in scale 1: 500 with respect to the official population data of the National Office of Statistics INE for the sake of numerical efficiency.

## Mobility between metapopulations and lockdowns

The trips between areas are data-informed from the aggregated mobile phone records in the first two weeks of March 2020. We have applied to them the same scale reduction factor as for the total population. Mobility is implemented stochastically. At each time-step agents located in population $i$ have a probability to travel to another population $j$ that is obtained by dividing the empirical trips by the local population of $i$. To keep the population constant in each area, each traveler from $i$ to $j$ is interchanged with a randomly selected agent in $j$, who travels in the opposite direction, from $j$ to $i$.

Regarding lockdowns, we are interested in controlling the number of seeds arriving at every sub-population. Therefore, we implement in the model two forms of lockdown. The first, named total lockdown, consists in completely blocking mobility in the entire system once the source region in which we have introduced the first cases (corresponding to the province of Madrid) reaches a certain number of cumulative cases, arbitrarily set at 2,000. This lockdown will be the standard unless otherwise mentioned. Such mechanism is thought to resemble what occurred in Spain and other European countries in the early stages of the pandemic. The number of local detected infections in Madrid reached 100 cumulative cases around March 6, the curve continued to grow in the next days with more cases being hospitalized, which eventually triggered the hard national lockdown one week later on March 14, 2020. In the model, we have arbitrarily set the threshold for the lockdown yet always allowing the epidemic to last long enough to generate measurable outbreaks in all the areas so that we can study the effects of multiseeding.

The model that we are running is stochastic and in the total lockdown the number of seeds reaching an area can vary between each realization. This is why we consider a second type of lockdown called *fixed-seeding*. This second type of lockdown is not applied at the same time in the whole country and it allows us to control the exact number of seeds entering an area. Essentially, we assign a number of seeds allowed to enter in each subpopulation. In fact, a common national threshold is fixed on the seeds divided by the local population of each destination area. The minimum threshold that we can set, and the one that we have used below, is 1/56 in such a way that even the least populated area of the country corresponding to Ceuta receives 3 seeds. The other areas will receive this quantity or more. At each time-step, we check for every agent traveling from any area $i$ to $j$ her status concerning the disease: if the

traveler is infected, the number of imported cases of population $j$ is updated, until reaching the established seed threshold for $j$. Once the threshold is attained, no further infected agents should be allowed to enter $j$. The mobility could have been maintained for non-infected agents, but as a simplification we stop all the trips to and from $j$.

## Mobility, population and epidemic data

The basic geographical units considered in this work are provinces and regions (maps can be seen in Fig A of S1 Text). They are selected due to the combined availability of epidemic and mobility data, and correspond to provinces (NUTS –European Nomenclature of territorial units for statistics– 3 areas) in Italy and Spain (where the islands have been aggregated at province level) and to regions (NUTS 2 areas) in England, France and Germany. The inter-area mobility has been obtained from anonymized and privacy-enhanced mobile phone location records of a telecommunication network activity by Kido Dynamics SA for Spain and from Cuebiq Inc. for England, Germany, France, Italy and Spain. Location data provided by Cuebiq are collected anonymously from opted-in users, who provided access to their location through a GDPR-compliant framework. In addition to de-identifying the data, the data provider applies privacy enhancements to preclude the re-identification of individual users. The operating system of the device (iOS or Android) combines various location data sources (e.g. GPS, WiFi networks, mobile network, beacons) to infer geographical coordinates. Several factors may affect location accuracy (which can also vary over time for the same device), but it can be as accurate as 10 meters. The data are extensively described in Ref. [38]. The Spanish mobile phone record dataset from Kido contains approximately 13 million devices with unique daily mobility patterns. The relative information regarding the sample extracted from Cuebiq data is available in Table 1.

The access to mobile phone data is fundamental to analyze and inform epidemic models to further design proper public health policies [51]. This is due to the large coverage of this type of data that accounts for all sort of mobility with high temporal and spatial resolutions. Such data sources demonstrated their potential in uncovering spatial heterogeneities of mobility responses across different scales [52–54]. Other alternative sources of mobility information, like census data, provide a static picture centered around commuting, which is relevant because of its recurrence, but that corresponds to around one half of the total mobility in cities (see, for instance, the mobility surveys in Barcelona and Madrid [55, 56]), and it may represent even less in middle to long range trips within a full country. In this case, having two datasets gives us the possibility of covering several European countries, and in the case of Spain where we have data from both to perform a stability analysis of the results.

The definition of mobility in the Spanish database from Kido refers to the concept of stay. Areas of residence are assigned at the province and regional level to the users according to the most common location of their stays outside office hours. If a device is observed outside the

**Table 1. Amounts of Cuebiq anonymous users and data points for each country under study.** Users and data points refer to the period of study from January 8 to April 14, 2020. K refer to thousands, M to millions.

| Country | Users | Data points |
|---------|-------|-------------|
| England | 474K | 412M |
| France | 245K | 162M |
| Germany | 220K | 216M |
| Italy | 174K | 236M |
| Spain | 200K | 109M |

residence area, then we consider that a stay has occurred for the day. These stays are aggregated to conform origin-destination matrices between provinces and regions. In the case of mobile phone records, inspired by differential privacy methods [57] and to avoid any potential disclosure of personally-identifiable mobility patterns, a small unbiased Laplacian noise with scale parameter of 5 (i.e., variance of 50 trips) is added to all aggregated values of stays. After adding the noise, any values below 10 are discarded from the sample as additional preventive measure. Such aggregated stays information are the basis for the mobility analysis presented in this work. This allows us to analyze the number of stays of residents of Madrid in the different provinces and, the other way around, the number of residents of those provinces that visited Madrid. Using trajectories of users from the Cuebiq data, we captured the movements between the geographic unit areas, creating daily origin-destination (OD) matrices for each of the five countries. The trajectories were aggregated over the unit areas, and discard the visits shorter than 1 hour (if a user is traveling, she can cross several areas without stopping). In all the cases and as above, we will take as basis the two weeks mobility until one week before the onsets in incidence in the different areas.

Regarding population counts and number of COVID-19 cases, the data has been collected from official public repositories.

## Data preparation

The focus is set on the first wave of the pandemic, until April 14, 2020. At this stage, we can assume the initial total population as susceptible, with one or a few clear centers radiating the spreading, which allows for a clearer spatial observation of the relevance of mobility and multi-seeding in the dynamics of the epidemics. The key metrics are defined from the incidence and mortality (number of deceases per day per capita) curves. These curves in the initial stages are affected by several issues like different protocols and rhythms of testing. It is easy, for instance, to detect the effect of weekends with a clear slowing down of testing and reporting. A process of smoothing is, hence, applied to obtain more reliable estimates on the epidemic trends. To do this, we take a running average over three days assigning the value to the central point. Once the curves have been smoothed-out (see Figs D-K of S1 Text), we record the magnitude of the peak and the local onsets (when the incidence is larger than a national threshold). The local onset provides the time window of analysis of the incoming mobility (three weeks to one week before). The list of national thresholds can be found at Table A of S1 Text. Tables B-F of S1 Text contain a detailed catalog of the onset and peak times, both for incidence and mortality, in all the areas considered.

On the mobility side, to see the effect of seeding, on the local outbreaks, for every geographic subdivision we consider only the incoming mobility per capita occurred between three weeks and one week before the local onset. For example, if the onset in area A occurred on March 21, 2020, the trips counted are those registered between March 1 and 14. We checked the correlations between the local epidemic peak and the mobility in different time ranges, but stopping one week before the local onset produced the best results. In this choice, we are taking the assumption that a week is needed on average to see the effects on any change in behavior on the epidemic curves.

## Epidemic epicenter indicator

To search for the country area which has most impulsed the first wave, what we will call the epidemic epicenter, the first step is to calculate for each area $i$ the Pearson correlation coefficient $R_i$ between the height of the incidence peak and the incoming mobility per capita from $i$.

Then we define a corrected geographic correlation score as

$$R_i^* = R_i \, \frac{n_i}{N_{max}}, \tag{1}$$

where $n_i$ is the number of unique destinations found in the data from $i$ and $N_{max}$ is the maximum number of destinations from any origin in the same country. $N_{max}$ coincides with the number of regions/provinces in the country minus one only if the origin $i$ connects to all the other areas. The areas maximizing $R^*$ are those most likely to have sustained the first wave in the country. This can be different from being the places that experienced the first cases, exporting cases is the first condition to cause a national-wide wave but it is also necessary that those seeds generate local outbreaks in the destination regions.

## Results

### Multi-seeding in a single population

First we gain insights into the role of multi-seeding in disease spreading using theoretical models and, later, we will focus on the empirical observations related to this phenomenon during the first wave of COVID-19 in Europe. For the sake of simplicity, we start the analysis with a single population where the seeds are the initially infected individuals in the simulation. In Fig 1, we show the curve of incidence (new daily infected cases and per capita) for the model as a function of time for a population slightly over twelve thousand individuals starting in each case with a given number of seeds. Only simulations in which the outbreak develops are considered. As can be seen in the incidence curves versus time (Fig 1A), multi-seeding in WM

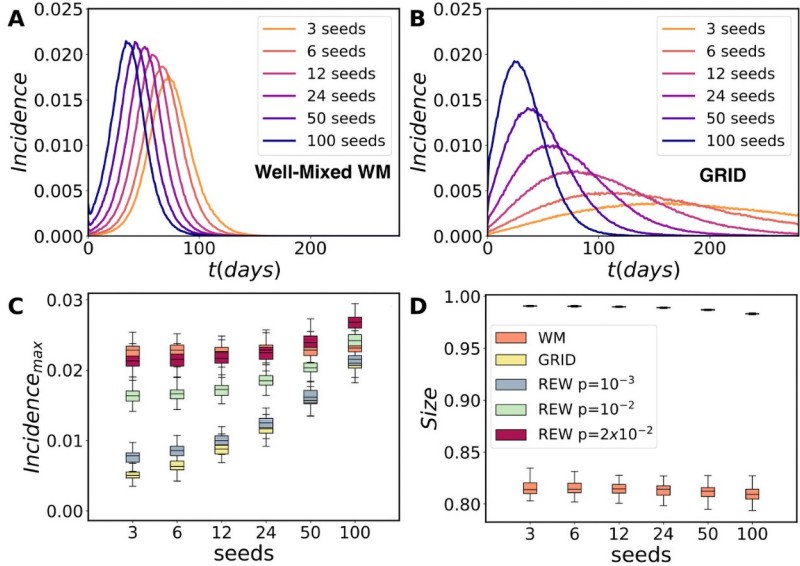

**Fig 1. Model and multi-seeding in a single population.** Average incidence curves versus number of seeds in a single population corresponding to a rescaled Madrid (12, 873 agents) with (A) the WM case and (B) the GRID topology. (C) Box-plot of the distribution of incidence peaks and (D) final epidemic sizes for a WM, the grid (GRID) and the rewired grid (REW) with different rewiring probability $p$ as a function of the number of seeds. The sizes for the GRID and REW appear together in the upper row of symbols, while the WM case is below alone. Note that the same increase of seeds in the grid contact network produces way faster and sharper peaks than the equivalent simulations for a WM population. One hundred simulations have been run for each scenario. Note that the size depends non-linearly on $\beta$. It was fit to produce sizes of the same order in the GRID and WM, but they do not exactly coincide so the height of the incidence peaks of REW with $p \to 1$ do not tend to those of WM.

essentially helps to accelerate the curves. In the case of the GRID contact network, the effect of multi-seeding is to accelerate the curves as well (Fig 1B), but the phenomenon is more dramatic. When the interpolating networks REW are considered, the height of the incidence peaks shows a weak growth with the initial seeds (Fig 1C), being almost logarithmic for the rewired networks and the WM models, and the total final size (fraction of individuals infected) does not seem to depend on the number of seeds (Fig 1D). Such independence occurs for high values of the infectivity rate $\beta$. This effect can be easily understood: More seeds implies that local outbreaks develop in different areas of the network and the global curve of incidence can attain larger values. However, in the long run the different outbreaks merge and the final size attained is essentially the same. On the other hand, with lower values of $\beta$, the size grows with the number of seeds (see Figs Q and T of S1 Text). This dependence is observable in the GRID, in which, the presence of bottlenecks can hinder the spreading of the disease. In the WM topology, the bottlenecks are not present and, therefore, the size does not depend on the number of seeds.

**Common non-linear features.**   Observing the regularity of the incidence curves of Fig 1A and 1B, one may wonder if there exist common features to both models and if it is possible to collapse the curves by normalizing time and incidence with a proper function of the seed number. As can be seen Fig 2A and 2B, it is indeed possible and there are some simple scaling arguments that explain how. For example, in a GRID contact network the contagions advance as wave fronts from the initial seeds. If these seeds have been placed at random, the average (characteristic) distance between them is given by $d_c \sim \sqrt{N/s}$, where $s$ is the number of seeds and $N$ is the population. The incidence peak is attained when the epidemic fronts cover $d_c/2$ and they all merge, after which the epidemic goes towards extinction. This implies that, since the front speed is fixed, the time for the peaks $t_{peaks} \sim (s/N)^{-1/2}$, which is confirmed by the fit to the simulation results of Fig 2C. The height of the incidence peaks follow the inverse relation with $(s/N)^{1/2}$. An interesting feature to note is that the important variable is the ratio between seeds and local population, we will numerically recover the same dependence in a metapopulation system.

The WM contact network is different because it is not static in time. However, model dynamics in these mean-field approaches can be approximated by those on a static small-world random network [58]. In one of this networks, as in the GRID the distance is measured by the number of links needed to connect two nodes through the shortest path. Assuming a tree-like local structure, the small-world property implies that the number of nodes reachable from an initial seed grows exponentially with the distance and the average distance between

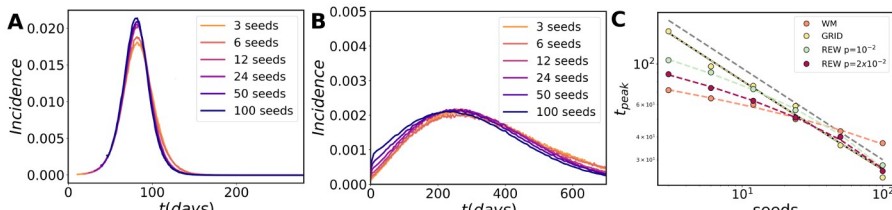

**Fig 2. Collapse of the epidemic curves in a single population.** The collapse of the epidemic curves produced by a different number of initial seeds in a WM population in (A) and in a population with a GRID contact network in (B). (C) The fit of the incidence peak time shows a different dependence on the number of initial seeds in the population. The peak time obeys to a power-law rule in the GRID scenario, while adding more and more rewiring probability to the contact topology leads to a logarithmic trend, which is fully recovered in the WM scenario. The incidence peaks behave similarly for all topologies and allows for the collapse of the epidemic curves in all the cases. The grey dashed line represents a power-law guide with exponent $\gamma = 1/2$.

seeds decreases as the logarithm of the inverse of $s$ [59]. To obtain the collapse, we need to subtract this logarithmic trend from the time scale, in the case of Fig 2B this means adding $10.2\log$ (seeds). The height of the peaks shows a similar logarithmic trend, in this case the collapse is obtained by subtracting $1.1410^{-5}\log$(seeds). Similar collapses are obtained for lower values of the infectivity parameter $\beta$ with the same functional forms ($\sqrt{s}$ for 2D GRID and logarithmic corrections for WM, see Fig W of S1 Text).

When we add a small probability of rewiring to the GRID to pass to a REW network, the system presents a cross-over between the GRID and the WM behavior (see the trends in Fig 2C). If $p$ is small, the network is locally a GRID and so, for large number of seeds (small $d_c$), the GRID scaling is recovered. In the opposite range, for small numbers of seeds, $d_c$ increases, the role of the long distance shortcuts becomes visible and the WM behavior dominates. If $p$ is big enough ($\sim 10^{-2}$), only the WM scaling remains. The most important question to notice is the strong non-linear relations that emerge between the number of seeds and the epidemic indicators in all the cases.

## A two populations model

In this first experiment, the seeds are effective since the beginning of the simulations and all of them start to propagate the disease simultaneously. A more realistic scenario with explicit importation of cases towards an area without active transmission is reported in Fig 3 by considering two subpopulations coupled by mobility fluxes. The spreading starts in one sub-

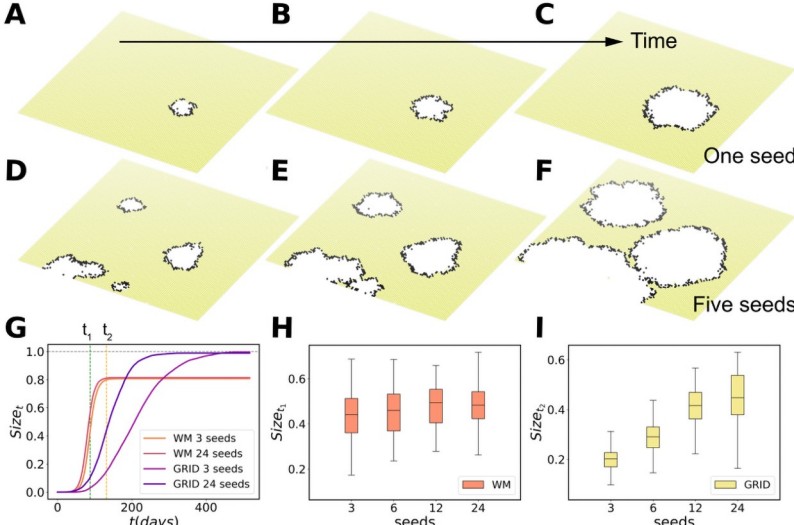

**Fig 3. Simulations for two populations.** The network now includes only two subpopulations, corresponding to rescaled version of Madrid and Barcelona with approximately 12873 and 11047 agents, respectively. The spreading is initiated at Madrid and the observations are performed at Barcelona, in this way the seeds in the second population arrive at a rhythm marked by the epidemic dynamics and the mobility between both cities. Once the number of infected travelers attains the threshold for the number of seeds all the trips are suspended and the epidemic evolution becomes internal to the subpopulations. In (A-F), the case in which the contact network is a GRID is represented to illustrate the impact of multi-seeding in a single realization. In the upper series (A-C), only one seed arrives. In the bottom row (D-F), five seeds arrive from Madrid. Both simulations are clones, the arrival of the first seed occurs at the same time and location in the grid. The graphical representation corresponds to the same times up and down: $t_0 = 66$ days in (A) and (D), $t_1 = 88$ in (B) and (E), and $t_2 = 132$ in (C) and (F). If a mitigation measure is implemented at a fixed time, e.g., $t_2$, the epidemic size is way larger in case of multi-seeding. In (G), we show the trend of the size as a function of time of a simulation with the WM and the grid contact topology with 3 and 24 seeds arriving from Madrid. Temporal evolution of the size at $t_1 = 88$ days for a WM population in (H) and for a gridded contact network in (I) at $t_2 = 132$ as a function of the number of seeds that arrive.

population with 10 seeds. We first simulate the spreading internally and then, at the end of each time step, the agents of area $i$ can travel to the other area. As an illustration of the effect of multi-seeding, Fig 3A–3F show the unfolding of the disease in the second sub-population for two replica simulations in which the contact network is a grid. Both rows are snapshots at fixed times in the simulation: $t_0 = 66$ days in Fig 3A and 3D, $t_1 = 88$ in Fig 3B and 3E, and $t_2 = 132$ in Fig 3C and 3F. In the upper row (Fig 3A–3C) only one infected individual is allowed to travel between the populations and to start a new outbreak, while in the bottom row (Fig 3D–3F) five seeds are allowed to arrive. As before, the spreading is way faster with larger number of seeds. If we allow the simulation to continue until the end, the final size does not depend on the number of seeds as occurred in a single population (Fig 3G). However, if the size is estimated at a given time, e.g. at $t_1$ or $t_2$ in Fig 3I and 3H, we observe a dependence of the cumulative number of infected individuals attained on the number of seeds, especially if the contact network has a spatial structure and it is not WM. This is important because if mitigation measures are taken at a certain time, both the maximum incidence (see Fig B of S1 Text) and the size do show a relation with the number of seeds (Fig 3I and 3H). Note that in the model lockdowns are never lifted in order to better study the effect of multiseeding, while this is an unrealistic scenario for the real world. In this sense, mitigation measures such as lockdown essentially slow down or stop the spreading process and, therefore, both the maximum incidence and the final size may depend on the number of seeds arriving and, ultimately, on the mobility between populations. If $\beta$ is lower, as occurred for a single population, it is possible to observe an increase of the final size with the number of seeds (Figs Q-W of S1 Text). The effect of stopping the simulation at an intermediate time is to foster this dependence.

## Multi-seeding in a heterogeneous metapopulation network

The next step is to study the role of multi-seeding phenomena on multiple structured sub-populations when heterogeneous populations and mobility flows are considered. The simulations start with 10 seeds in a single sub-population and the epidemic curves are then observed in every area. The peak of incidence is displayed as a function of the incoming trips per capita in Fig 4. The incoming trips are proportional to the seeds and the division by the destination population is important because three seeds, for example, have a much larger impact in a small population than in a large one, whereas to get a similar outbreak each area must import a number of seeds proportional to its population (see Fig C of S1 Text). Each sub-population corresponds to a symbol in the plot of Fig 4. The mobility and population heterogeneity translates into fluctuations in the height of the incidence peaks. To quantify the relation between peak of incidence and mobility, we measure the dispersion of the peak values across areas $\sigma$, which is the standard deviation for the peaks height. When $\sigma$ is low, no appreciable differences can be observed among the incidence peaks of geographical locations. Additionally, we fit the plots with the LOESS regression method, which yields a $R_L^2$ that informs on the quality of the fit and the relation. The reason to use LOESS rather than, for instance, Pearson correlation is its versatility, since being a local and non-parametric polynomial method it does not introduce a priori assumptions on the functional form for the relation between mobility and epidemic indicators.

The first simulations in Fig 4A and 4B consider a scenario in which only a small number of infected individuals (seeds) are allowed to travel. A threshold is imposed on the maximum number of incoming seeds per capita allowed to enter in each subpopulation. Note that the curves are nearly flat, the values of $R_L^2$ are very low as are as well those of $\sigma$. This is natural since in this scenario after the first importation of cases the outbreaks are developed locally. After the seeds arrive, traveling to and from that sub-population is no longer permitted. Next,

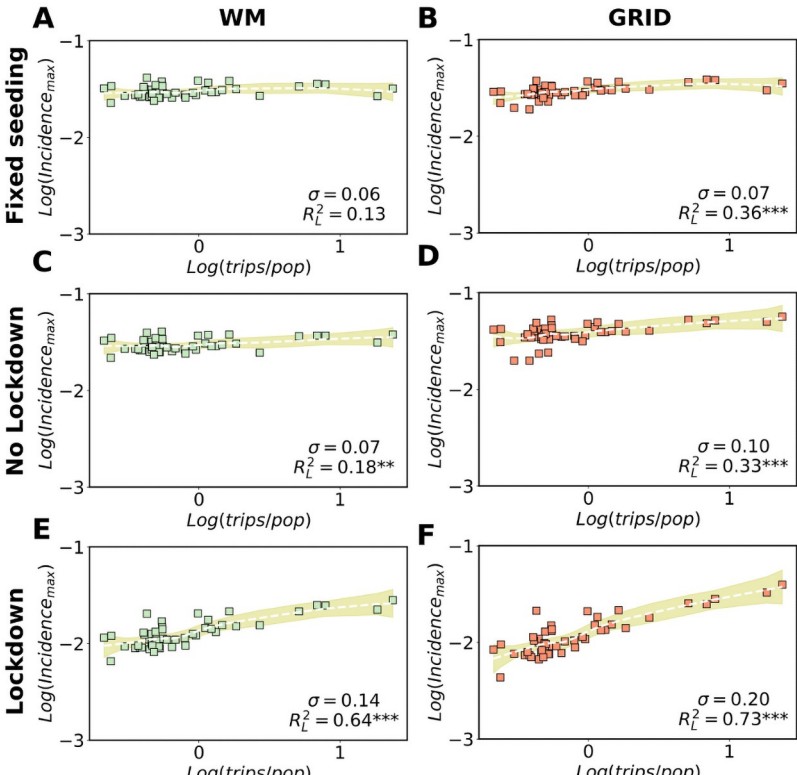

**Fig 4. Models with heterogeneous populations.** Comparison of the effects of confinement policies, seeding and topology on the correlation between incidence peaks and mobility from the source (seeding) in a metapopulation model. In all the plots the height of the incidence peaks are displayed as a function of the incoming trips in every area divided by the local population. In (A), simulation of WM populations in each province allowing to travel only a fixed number of seeds per capita from the source (Madrid). The threshold is the destination population divided by 56 so that at least three seeds can travel in the least populated area. In (B), the same for gridded contact networks in the provinces. Both scenarios are again considered in (C) and (D), but this time without constraints in the travels between provinces and allowing thus multi-seeding. Finally, in (E) and (F), the simulations are repeated with a national lockdown applied when Madrid arrives at 2000 cumulative cases. All results are averaged over 100 simulations with each square corresponding to the average value obtained in a province. In all cases, curves and fits are obtained using the non parametric LOESS method, as also the values of $R_L^2$ corresponding to the R-squared of fit and data. The shaded areas correspond to the 95% confidence interval. $\sigma$ is the standard deviation and represents the dispersion in the values of the peak of incidence across provinces.

Fig 4C and 4D displays a no-intervention scenario, hence we fully allow multi-seeding. Here multi-seeding is occurring but its effect is covered by the absence of any containment policy, $R_L^2$ slightly increases and the same can be said for $\sigma$. A gridded contact network favor the effect of multi-seeding but the range of values of the three indicators are still similar to those of the fixed number of seeds. Finally, we consider the case in which a system-wide lockdown is applied. The effect of the lockdown is simulated by decreasing the infectivity rate $\beta$ by one half, and ceasing all inter-area mobility. We do this in order to model and measure the most extreme and direct effects of a lockdown on the seeding of local areas. As reported in Fig 4E and 4F, the combination of both multi-seeding and lockdown can increase at least in one order of magnitude the variability of the incidence peaks with respect to the incoming trips. Indeed, the variance explained, $R_L^2$, now multiplies by a factor 5. This phenomenon has a simple explanation: multi-seeding from the geographical source is stronger with larger mobility from and to it. Every seed that arrives from different areas of the network has the potential to start an outbreak, with the overall epidemic curve growing faster in the places where more

seeds have been imported. Without lockdown, the epidemic continues the progress locally in every area and the role of the seeds from the original source diminishes. On one hand, the local epidemic curves develop and, on the other one, seeds start to arrive from other origins. Here, the signal originally produced by the mobility from the epidemic epicenter is partially lost because of the interfering mobility from the other seeding sources. New outbreaks in the destination communities are not exclusively caused by the epidemic epicenter. In contrast if a lockdown is applied, the progression of the disease slows down and the height of the incidence peaks reflects the number of seeds received in every area from the source at that moment. With lower $\beta$, we find as well a stronger dependence between the maximum incidence, the final size and the incoming trips per capita if both factors, multiseeding and lockdown, are present than if only one is considered (see Figs R and U of S1 Text).

## Effects of multi-seeding in the first pandemic wave

Once characterized the role of multi-seeding in models, we check whether its effect can be observed in empirical data. We start our analyses by showing the connection between mobility from the region or province of the epidemic epicenter to every destination and epidemic features such as the maximum local incidence or mortality in the area of destination.

In Fig 5, we show the correlation between the height of the incidence peak and the incoming trips per capita in the local population taking as the epidemic epicenters: (A) Lancashire for England, (B) Champagne-Ardenne for France, (C) Munich for Germany, (D) Milan for Italy and (E) Madrid for Spain. Note that the regions can have sometimes the same name as their central cities, but they are more extensive. These regions have been selected as the epidemic epicenters because they maximize $R_i^*$. The first question to notice is that the correlation level is not uniform across countries. All the $R_L^2$ values are significant at 5% level, but in England the correlation incidence peak is low, $R_L^2 = 0.38$, while in the other countries it is higher. One of the assumptions of this analysis is that there has been a single importation zone as source and this may not have been the case in England. Another issue to take into account is that France, Italy and Spain imposed a very severe mobility reduction [38, 60, 61], while others operated a lighter reduction of total mobility [62, 63]. Moreover, in Spain and Italy, the focus of the spreading was registered in the largest mobility hubs of the country, i.e., Madrid and

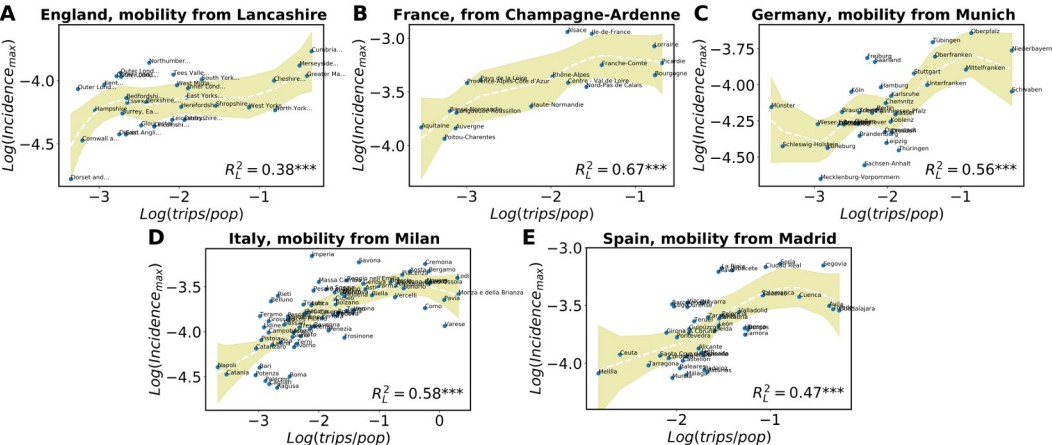

**Fig 5. Correlations of incidence peaks and mobility.** Peaks of incidence versus total trips per local capita received from (A) Lancashire, (B) Champagne-Ardenne, (C) Munich, (D) Milan, (E) Madrid, during two weeks until one week before the local onset. In all cases, the curves with the fits are obtained using the non parametric LOESS method, as also the values of $R_L^2$. The shaded areas correspond to the 95% confidence interval. The three stars next to the $R_L^2$ values indicate a $p$-value below 0.01.

Milan, which intuitively may have fostered the spreading. The same cannot be said for France and England, where London and Paris did not played a critical role. Lastly, in Germany Munich seem to have participated in the spreading of the virus, which is indeed a mobility hub of the country, whereas other important hubs such as Köln did not exhibit a similar behavior.

The picture is maintained if one considers the mortality instead of the incidence, as shown in Fig 6. The mobility for the analysis is the same as the one considered for the incidence but the mortality curves need a longer time to peak than the incidence (see Figs I-K of S1 Text). In general, the higher the mobility between the country areas and the supposed source (Champagne-Ardenne, Munich and Madrid), the faster cases are imported and the more deceases are observed per capita. This further contributes to the hypothesis that the arrival of multiple-seeds increases the local speed of the epidemic and, consequently, the rhythm at which mortality locally grows before mitigation measures are taken to prevent the expansion of the disease.

Using $R_i^*$, we explore in Fig 7 the correlations between incidence peaks and mobility from different origin regions. It is important to stress that this method helps to identify which regions fit best as impulsors of the local outbreaks of the rest of country areas. These epidemic epicenters are not necessarily the areas with the first detected cases, but usually correspond to those joining mobility centrality and early development of local outbreaks. As we can see, the areas that concentrate the highest values of correlations are usually clustered in space with the only exception of Madrid. This is not fully a surprise, since the Spanish infrastructures are traditionally organized in a radiant way and Madrid act as the main mobility center of all the country (Fig 7E). The epidemic arrived at different cities of Spain in February, but Madrid was the area in which a local outbreak developed fastest in late February-early March and spread to the rest of the country [64]. By only correlating incidence and mobility, the heat map in Fig 7 actually confirms such scenario. Similarly, the epidemic in Italy was detected initially in more than one focus in Lombardy and Veneto in the North of the country and extended from there in the first wave [65, 66]. The province of Milan (capital of Lombardy) shows the highest value of $R_i^*$ (darkest blue color) followed by Bergamo in Lombardy as well (Fig 7D). Germany has several mobility hubs, including Berlin, Hamburg, Frankfurt, etc. However, the geographical spreading of the pandemic matches our map with an initial focus on the South, in Bavaria, and around Munich (Fig 7C), confirming a recent study on the first outbreak in the country [67]. In the case of France, the main mobility hub is Paris and the region around it (Île de France), but, coinciding with our map, the disease entered through the regions in the North-East (Fig 7B) and spread to the rest of the country, in agreement with a recent geo-epidemiological study [68]. The same can be said for England, where London did not participate much in the spreading despite being the main mobility hub, not as much as the regions in the North (Fig 7A). Here, the National Office for Statistics highlights the North-West region as one of the epidemic clusters of England in mid March [69]. Hence, in general we can define the epidemic epicenter as the area $i$ that maximizes the corrected correlation score $R_i^*$. It can be concentrated in a single geographical division or shared by several neighboring areas, and, most importantly, it does not necessarily coincide with the capital or the main mobility hub of the country. Recall that the mobility data for all the countries was retrieved from GPS traces except for Spain that came from mobile phone records. This is the only country where we observe a concentrated maximum of $R_i^*$, we repeat, therefore, the analysis with GPS data as a sanity check in Fig L of S1 Text. The results are quite similar, even though noisier due to the way smaller population sampling but this confirms the robustness of our results. By this analysis, we can understand that those areas with higher peaks of incidence and mortality are characterized by more mobility per capita from/to the epidemic epicenter.

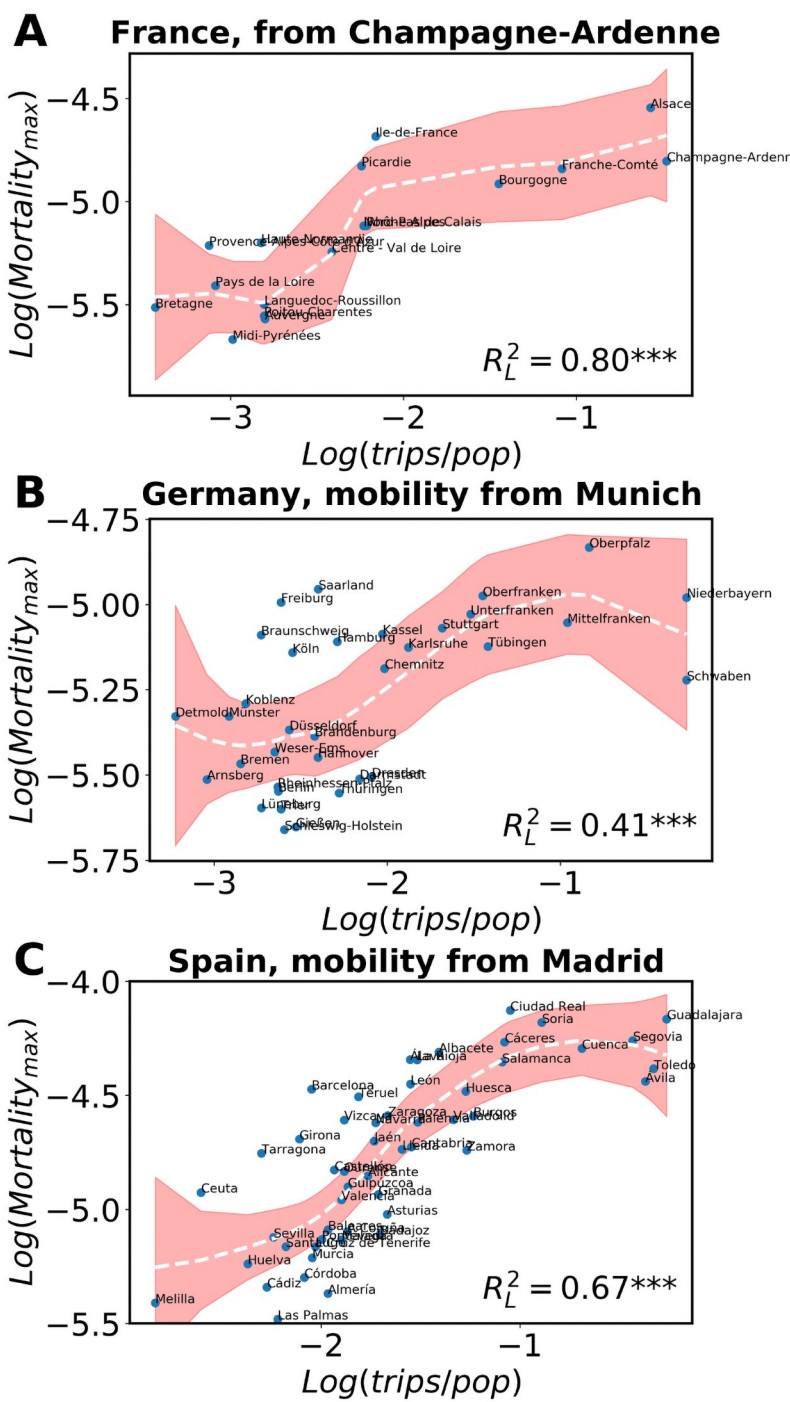

**Fig 6. Correlations for mortality versus mobility.** Peak of mortality (where available) versus total trips per local capita from (A) Champagne-Ardenne, (B) Munich, (C) Madrid, during two weeks until one week before the local incidence onset. In all cases, the curves with the fits are obtained using the non parametric LOESS method, as also the values of $R_L^2$. The shaded areas correspond to the 95% confidence interval. The three stars next to the $R_L^2$ values indicate a $p$-value below 0.01. The origin of the trips are the areas whose $R^*$ was highest in each country.

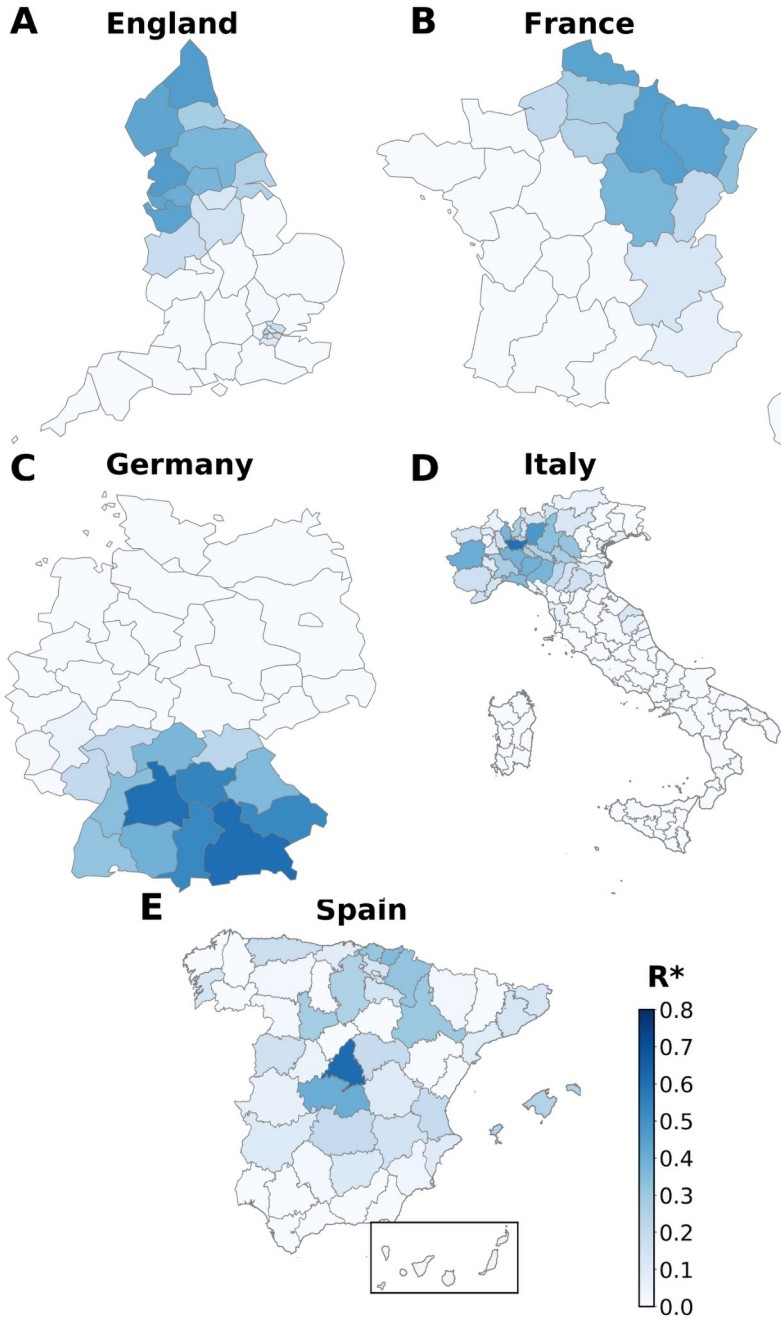

**Fig 7. Epidemic epicenter analysis.** For each origin of each country we check the corrected R-squared $R^*$ correlation of mobility and destinations incidence peaks. Darker areas represent those where most probably the spread was originated from. Administrative boundary data were obtained from GADM for Italy and Spain (https://gadm.org), and from EuroStat (https://ec.europa.eu/eurostat) for the rest of countries.

To further understand the relevant variables contributing to the heights of the incidence and mortality peaks, we perform a linear multivariate analysis including five variables: logarithm of two weeks total trips per capita (in destination) until one week before the incidence onset, population density, distance from the epidemic source (the area maximizing $R_i^*$), population of every unit area and local onset times. Note that the onset time is an intrinsic epidemic variable and as

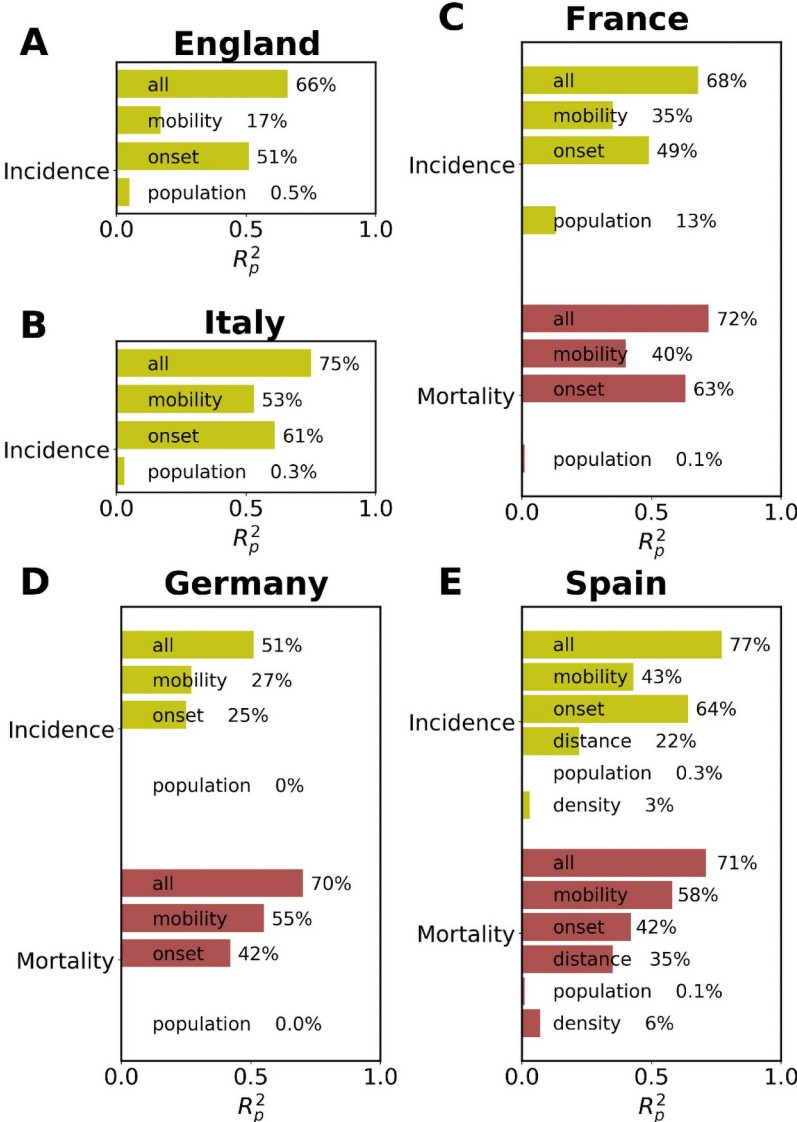

**Fig 8. Multivariate analysis.** Bar chart with the multivariate correlation results in $R_p^2$ for the incidence and mortality peak (where available) heights in each unit area as a function of the mobility per capita and onset times. In every case, the upper bar corresponds to the simple correlation with all variables, mobility, onset time, population and population density where available. In this analysis, $R_p^2$ refers to R-squared of the linear Pearson correlation.

such it can depend on the others (e.g., mobility). For those few areas that did not reach the established onset and to keep them in the analysis, we have taken the time of the maximum incidence peak as the onset. As can be seen in Fig 8, the mobility from/to the source is the main variable after the onset to explain the variance in the peaks and it scores much better than the the density of the local population and the distance to the source. Curiously, in the countries where we have mortality data, the variance explained by mobility is larger for the mortality peaks than in the incidence ones. This may be explained by the variability in the test policies in the early stages, while the deceases are better documented, the new cases per day officially recorded represented only a partial view of the situation. By adding all the variables, the explained variance increases over 50% in all the countries and even it is close or over 70% in most of them.

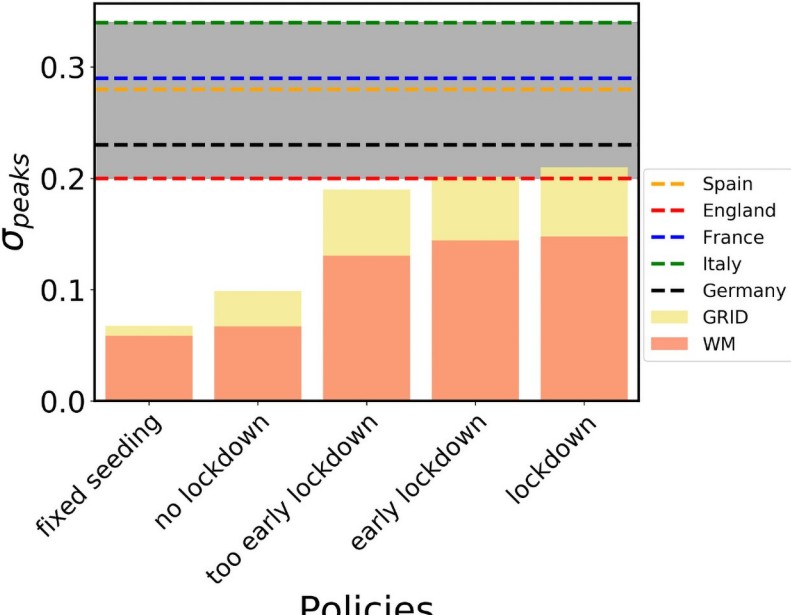

**Fig 9. Policy comparison.** Incidence peaks standard deviation in for the multipopulation system with different interventions scenarios. The standard deviation is measured in plots similar to those shown in Fig 5 both for empirical data and for the model simulations. The number of seeds in the empirical data is related to the mobility, while in the fixed seeding simulations we let each subpopulation reach a common threshold of cumulative incidence before closing the province borders. No lockdown is the scenario with no interventions. Too early, early and normal lockdown corresponds to a national lockdowns issued when Madrid (the source) reaches 1000, 2000 and 3000 total cases, respectively. After the lockdown $\beta$ gets halved and all trips across borders stop.

As mentioned before, the incidence data in the early times of the pandemic suffers from some uncertainties. To check the quality of our trends, we have repeated the analysis in Spain with results of the national seroprevalence test [70]. The Spanish Ministry of Health released the first results of a study on the prevalence of antibody response in the population (https://www.mscbs.gob.es/gabinetePrensa/notaPrensa/pdf/13.05130520204528614.pdf). The analysis tested 60, 983 individuals from different provinces, age groups and socio-economic contexts in a sample obtained with the same mechanisms as the census. The results are thus expected to be representative of the extension of the disease in the population (the size at the time in which the test was performed). Repeating the analysis, the mobility in terms of stays per capita explains 64% of the variance of the antibody prevalence ($R_P^2 = 0.64$), while the full multivariate model accounts for 68% and the comparison between the statistical model and the data yields $R_P^2 = 0.68$ (see Figs O and P of S1 Text). Despite the data of the prevalence analysis has been obtained much later than the peaks, these results strongly confirm the scenario we have found of dependence between the peaks (mortality and incidence) and the mobility.

Finally, in Fig 9, we perform an analysis of the effect of different types of traveling restrictions on the relation between the incidence peaks and the arriving seeds. The gray background marks the area of the empirical values observed for $\sigma$ in different countries, while the color bars portray the models outcome in Spain. Regardless of the contact network topology, the model is not able to attain the empirical zone unless both multi-seeding and lockdown are present. Given that the incidence peak can be reached after the lockdown, different times for the travel restrictions modulate the values of $\sigma$. Still, modifying the lockdown times one can obtain values of the relations that agree with the order of magnitude of the empirical observations. As it is observable in the case with lower $\beta$ (see Figs S and V of S1 Text) the values of $\sigma$

are stable and consistent with the empirical ones, while the lockdown times in the WM scenario highlight a vanishing effect of lockdowns as they are introduced later in time, confirming our hypotheses on the role of late interventions. In all cases, it is necessary the concurrence of both mechanisms of multiseeding and lockdowns to reproduce the empirical values of $\sigma$.

## Conclusions

Understanding the impact of mobility on the early stages of spreading of an epidemic is the key to design efficient public health responses. In this work, we address the question of how relevant is multi-seeding for an epidemic. Beyond the classical analysis centered on the arrival time of the first imported cases, here we focus on the effect that the incoming of more seeds have on the evolution of the local epidemic curves. We start with a theoretical analysis by building a SEIIR metapopulation model with different topologies for the contact network inside each population (including well mixed populations WM, a GRID and intermediate networks called REW).

In a single population, we can already observe quicker outbreaks and higher incidence peaks with more seeds, although the final epidemic size does not seem to depend substantially on the seeds. This relation is weaker for a WM population than for other contact networks. Interestingly, simple scaling arguments based on the distance between seeds in the contact network allow us to collapse the incidence curves leading to a single functional form. The particular scaling relations vary with the dimensionality of the contact network, but in all the cases the connection between the time for the peaks or the height of the incidence peak is highly nonlinear. The impact of multi-seeding can be further illustrated by observing the phenomenon in a two populations system, assuming a contact network in a GRID configuration and monitoring the spreading in time. More seeds means an increase in the number of independent outbreaks and a much faster spreading in the system. This naturally leads to higher incidence. If one waits until the spreading is over, the final size is similar regardless of the seeds, but if an intervention is going to be taken at a certain time, as a lockdown, the cumulative number of cases is much larger in case of multi-seeding. Indeed, in the model with many populations, the number of seeds that arrive from the source is related to the number of trips from the geographical source of the disease. As an important question, we find that the important variable that controls the multiseeding effect is the number of seeds arrived over the local population. A connection between the peaks of incidence and the mobility is observed in all the cases, but it is weak if no mitigation measures are taken. On the other hand, the application of lockdowns enhances the spatial differences in seeding and, consequently, on epidemic indicators. We have used a relatively high value of $\beta$ to produce a final epidemic size over 0.8. If lower values of $\beta$ are used, i.e., to model less infectious diseases, the main results are confirmed. There is a difference in the behavior of the final size, which now can depend on the number of seeds. Still, the effect of mixing multiseeding and lockdowns is to strengthen these relations. Such mixture is necessary to recover the empirically observed measures.

These non linear relations between epidemic curves and seeds are our main theoretical results. To test if these results are consistent with a real case, we performed a simple correlation analysis in the context of the present COVID-19 pandemic in England, France, Germany, Italy and Spain. Despite the lower testing rates of the first wave, we find a clear relation between the mobility from/to Madrid, Milan, Munich, Champagne-Ardenne and Lancashire for each country taken into account and the heights of the peaks of incidence and mortality of the relative national subdivisions. Additionally, we are able to uncover back in time the most likely geographical source of the spreading of the disease by a direct correlation analysis, recovering the pathways according to the history recognized by the local health authorities.

The main contribution of our work stands on two plains: on one hand, we offer empirical evidence of connections between multi-seeding and the severity of the epidemic at the population level on different countries that enforced different confinement measures. On the other hand, we provide a theoretical explanation of such relation by implementing a SEIIR model. The model shows the effects of mobility fostering multi-seeding, more visible thanks to the implementation of lockdowns and further enhanced if the contact network has a spatial structure. We hope that this study will highlight the importance of mobility in an epidemic situation, which goes beyond a first direct relation between arrival times and inflow of trips, and help stakeholders and decision-makers to design more efficient responses. Especially, it must be taken into account that mitigation measures such as lockdown will induce a strong relation between mobility from/to the source and cases. The areas with strong mobility from the source with low local populations require additional attention since they are likely to develop more violent outbreaks. In contrast, reducing the seeds from the source help to generate slower outbreaks with lower peaks, which are more manageable by the health system.

## Supporting information

**S1 Text. This document includes in a single pdf tables with further data descriptions and figures with extra simulation results to sustain the generality of our findings for other model parameters.**
(PDF)

## Author Contributions

**Conceptualization:** Mattia Mazzoli, Alberto Hernando, Sandro Meloni, José J. Ramasco.

**Data curation:** Mattia Mazzoli, Emanuele Pepe, David Mateo, Alberto Hernando.

**Formal analysis:** Mattia Mazzoli, Emanuele Pepe, David Mateo, Alberto Hernando, Sandro Meloni, José J. Ramasco.

**Funding acquisition:** Ciro Cattuto, Paolo Bajardi, Sandro Meloni, José J. Ramasco.

**Investigation:** Mattia Mazzoli, Emanuele Pepe, David Mateo, Ciro Cattuto, Laetitia Gauvin, Paolo Bajardi, Michele Tizzoni, Alberto Hernando, Sandro Meloni, José J. Ramasco.

**Methodology:** Mattia Mazzoli, Emanuele Pepe, David Mateo, Ciro Cattuto, Laetitia Gauvin, Paolo Bajardi, Michele Tizzoni, Alberto Hernando, Sandro Meloni, José J. Ramasco.

**Project administration:** José J. Ramasco.

**Supervision:** Sandro Meloni, José J. Ramasco.

**Validation:** Mattia Mazzoli, José J. Ramasco.

**Visualization:** Mattia Mazzoli.

**Writing – original draft:** Mattia Mazzoli, Emanuele Pepe, David Mateo, Ciro Cattuto, Laetitia Gauvin, Paolo Bajardi, Michele Tizzoni, Alberto Hernando, Sandro Meloni, José J. Ramasco.

**Writing – review & editing:** Mattia Mazzoli, Emanuele Pepe, David Mateo, Ciro Cattuto, Laetitia Gauvin, Paolo Bajardi, Michele Tizzoni, Alberto Hernando, Sandro Meloni, José J. Ramasco.

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
