## [Decision Letter · Decision Letter 0]

26 May 2021

Dear Dr Ramasco,

Thank you very much for submitting your manuscript "Interplay between mobility, multi-seeding and lockdowns shapes COVID-19 local impact" for consideration at PLOS Computational Biology.

As with all papers reviewed by the journal, your manuscript was reviewed by members of the editorial board and by several independent reviewers. In light of the reviews (below this email), we would like to invite the resubmission of a significantly-revised version that takes into account the reviewers' comments.

We cannot make any decision about publication until we have seen the revised manuscript and your response to the reviewers' comments. Your revised manuscript is also likely to be sent to reviewers for further evaluation.

Sincerely,

Alex Perkins

Associate Editor

PLOS Computational Biology

Tom Britton

Deputy Editor

PLOS Computational Biology

Reviewer's Responses to Questions

**Comments to the Authors:**

Reviewer #1: In their current manuscript, the authors investigate spreading dynamics described by a compartmental model on a network. In particular, they consider the impact of simultaneous importation (seeding) of infected and the implications of non-pharmaceutical interventions such as lockdowns and travel restrictions. The study is inspired by the current SARS-CoV-2 pandemic. The underlying network that forms the substrate of interactions is derived from mobility data (mobile-phone traces) of 5 European countries: Spain, England, France, Germany, and Italy. The main finding is that parallel seeding and interventions lead to spatially heterogeneous outbreak patterns (opposed to a diffusion wave). The study focuses on the period of the first wave in spring 2020.

The considered model of the spreading is taken from previous literature. Therefore, the novelty arises from the inclusion of empirical data and multiple importations. Due to modern mobility behavior, countries are closely linked to each other and any globally acting virus will have many different entry points into a country. The main points of my criticism concern the following questions: Why now? How innovative?

Why now?

The period considered in the study is spring 2020, when the first wave of the current pandemic occurred. Since then, there have been 2 or more additional waves. That triggers the question how universal the findings are. Similarly, how relevant are the findings in light of vaccination campaigns that have gathered speed in the countries considered?

How innovative?

The observation that an increased number of importations of infected leads to increased numbers of infected is almost self-explanatory. Every infected arriving at a susceptible part of populations acts as an independent nucleation core of a local outbreak.

Concerning the inclusion of mobile-phone data, I agree that these data sets have a lot to offer in terms of temporal and spatial granularity. On the other hand, what is the advantage compared to census data from pre-COVID times adjusted for movement restrictions (or other data sets such as Google mobility)? Related to the mobility, the finding seems to be that higher levels of mobility result in stronger transmission. In words of the conclusion (l.422/423): “Lockdowns slow down the progression of the disease”. Isn’t this intuitive?

Some comments on the presentation and additional questions:

1. Line 59 (gridded contact): Can the authors elaborated on their idea of a gridded contact network? Is it a regular square lattice (discretisation of a 2D space)? Is this a realistic assumption for a contact network in an urban area?

2. The scaling of 1:500 (l.66) is not in line with the number of agents representing Madrid (12k) and Barcelona (11k).

3. Line 82/83, 366 Remove URL from the main text

4. The text of lines 95 to 99 would be more accessible as a table.

5. Line .108: If the Laplacian noise is applied to randomize the exact location, the scaling parameter should have units, e.g., 5km.

6. The statements in lines 104/105 and lines 119/120 are redundant.

7. Some sentences read strange/colloquial/too complicated or are not needed for the flow of the text. See, for instance,

i. lines 133/134:

“The most straightforward one is the standard in this metapopulation modeling framework and it is the so called well-mixed (WM) population”.

ii. line 140: “and, therefore, one may wonder what occurs when the space enters into the equation.”

iii. line 218: “The effect as can be seen is minimal.”

iv. line 235: “the mobility with the initial source stops being so important”

I suggest a thorough proof-reading of the text to check the wording.

8. The abbreviation WM (well-mixed) is introduced in line 134 and should be used consistently for the remainder of the text.

9. Fig.1:

i. panel (c) The idea of a small-world network (rewired grid) is interesting. As more shortcuts are introduced, the more the peak incidences resemble the WM case (up to 50 seeds). It should be discussed why the REW leads to higher peaks compared to the WM model for 100 seeds. Is this because the existence of hubs facilitates the spread?

ii. Caption: The statement “The peaks and the sizes of the box plots are collected from the single simulation realizations and do not necessarily match those of the averaged curves.” can be safely removed. It is clear that box plots show the statistics of an ensemble of simulations.

10. Lines 169/170: What is the effect of the exchange of agents? Could there be an echo effect, that is, movement of infected to the location of origin of agent i? To keep the population constant, the authors could also assume a commuter model, where every agent returns to their home location.

11. Fig.2g:

1. Why does the WM curves do not reach a level of 1 as in the GRID cases, but saturate at a lower level? The same question applies to Fig.1d.

2. Can the curves be collapsed into one by rescaling, e.g., by the number of seeds or number of contacts? This question also applies to Figs.1a-b. This might give a handle on the mechanism and predictable impact of containment policies.

12. Lines 188/189: Lockdown measures might slow down the spreading, but they only stop the process if the number of infected reaches zero. With a finite residue, there will be an upsurge the moment they are relaxed or lifted.

13. Lines 212/213 (and line 180): I do not understand the notion “allowed to travel” and “allowed to enter”. Either the process of mobility is stochastic or there is a fixed number of seeds. The authors should clarify how the mobility works.

14. Fig.3:

i. What is the reason for dividing by 56? How does this related to three seeds?

ii. The authors should try to find a different display of the data points. The log-scale for the y-axis is of little use for almost horizontal alignment of points. If that was the main information of the plots, a table stating the sigma and R^2 values would be more suitable.

15. Line 223: How are the assumptions of “beta by one half, and ceasing all inter-area mobility” justified? Is the former backed up by empirical evidence on social distancing? Concerning the latter, there is surely an element of non-compliance/essential travel.

16. Line 271/292-297: The “origin zone” can also be found looking at the history of the primary cases in a country, that is, the start of the transmission chains (arrival from travelers from China, physical proximity to other countries with earlier outbreaks). I do not understand the presented reasoning. Once the coronavirus arrives in a country, it spreads rapidly to all areas. To trace the point of origin, the locations/nodes of underlying network could be rescaled using a network distance. Cf. Ref.[31].

17. Lines 447-453: I would have liked to see more on the impact of regional lockdowns considered in the simulations of the current study (cf. Fig.8 for Madrid). This would be insightful for more targeted intervention strategies in the future.

To sum up, while the implementation of the methodology of the study looks sound from a technical point of view, but there are open questions on the parameter selection and network realization. More severely, I fail to see the key innovation of the investigation at the moment. However, I would invite the authors to clarify the scope, key findings, and purpose of the study in a revision. This could include, for instance, a quantitative analysis of the impact of different levels of lockdown: a reduction of X percent in mobility delays the peak by a factor of Y. At this stage, too many things are unclear or seem to be intuitive and of qualitative nature.

Reviewer #2: In this manuscript the authors investigate the role played by human mobility in shaping the heterogeneity observed in the disease incidence across different locations, showing how mobility patterns could become especially relevant in presence of strong restrictions on individual movements and contacts.

COVID-19 is used as an illustrative example for the carried out analysis.

The manuscript relies on multiple approaches. The main one consists of a modeling analysis based on a meta-population SEIIR model, accounting for multi-seeding (multiple introduction of the infection in a population) and realistic mobility patterns derived from digital data records (GPS records from the use of mobile phones). The model is used to highlight that multi-seeding and a spatially structured contact network may explain the geographical heterogeneity observed in COVID-19 incidence and mortality in five European countries. The analysis suggests that highly connected regions are more likely to experience larger outbreaks before control interventions are implemented.

Statistical analyses are also provided using real COVID-19 data retrieved from different countries to highlight the relation between the mobility patterns and the data on mortality/incidence peaks.

I personally find this article technically sound and quite interesting, providing new insights to the spread of epidemics with potential implications for public health policies. Specifically, I really enjoyed the approach adopted by authors to identify the geographical area that likely acted as the initial source of infection across different countries. As such, I recommend the paper for publication in PLOSCB, after few but essential revisions.

My main concern regards how the methods are described in the main text.

In particular, a couple of assumptions are not sufficiently clear to me and some technical details are reported in results instead of in the methods, making it a bit hard-working to assess the appropriateness of specific analyses and hindering the overall understanding of the paper. For instance, I didn't get how and why different digital data were used for the different countries (Kido for Spain and Cuebiq for others?), why 4 contacts per individual were assumed in the simulation, and how the timing for the lockdown was assumed/simulated. I feel that these assumptions do not strongly affect the general findings of the paper, but they should be better specified in the text.

I also suggest the authors to provide an overview of the rationale of the entire analysis at the beginning of the methods, highlighting the different approaches and data used for this study.

Minor comments:

I suggest to use the term "homogeneous mixing" when defining the "well mixed" network

I assume that the "GRID" notation that appears in Figure 1 refers to the "lattice network" configuration mentioned in the text. Please use the same term for the text and figures. The authors also say “… a lattice as contact network (see below for details)” but no details are available on this in the main text or in the Supplement.

When considering a lattice structure for the network, the authors state that "every individual is a node and can interact only with her nearest neighbors". Do you mean that, under this assumption, individuals have only contacts with those living in adjacent cells on a grid representing different provinces? Please clarify the point.

A better description of what sigma represents (standard deviation?) and how it could be interpreted would be helpful (e.g. when sigma is low, no marked differences could be appreciated between different geographical locations)

Fig 1:

- In panels c) and d) the order of the legend is a bit confusing. For the three rewired network models (REW), the probabilities assigned are not in increasing nor decreasing order. Would be better to put in order the three REW (p=2x10**-2, p = 10**-2 and p = 10**-3).

- Panel d) is hardly visible.

Caption of Fig 2: “a function of time step" should read "a function of time"

Caption of Fig 3:

- “The threshold is the destination population divided by 56 so that at least three seeds can travel.” Why this threshold? Where is the relation between the 56 factor and the 3 seeds? Explain better.

- “Finally, in panels e and f, the simulations are repeated with a national lockdown applied when Madrid arrives at 2000 cases.” I guess the 2000 cases are cumulative. Please clarify.

Finally, please note that Figures S4-S11 from the supplementary material have all the panels with different ranges for both x and y axis, which hinders the comparison between them.

Reviewer #3: The manuscript provides an analysis of the spatial spread of the COVID-19 epidemic in 5 countries in Europe during spring 2020. Focus of the analysis is the role of multi-seeding coupled with social distancing on the spatial heterogeneities in incidence and mortality.

The study leverages on high resolution mobility data collected by means of mobile devices. The analysis of the COVID-19 epidemic is complemented by the analysis of simplified synthetic spatial systems to better highlight the role of the different ingredients (multi-seeding, local network of contacts, timing of the intervention).

The study tackles an important fundamental question, and the case study of the COVID-19 epidemic is highly relevant. The analysis presented provides important insights on the early-stage spatial propagation of COVID-19 in the European countries considered. On this basis, I feel that the manuscript has the potential to provide a nice contribution to PLOS Computational Biology. At the same time, however, the writing and the presentation of the work is unclear in many parts. With this respect, I believe that there are some major points that would need to be addressed.

The manuscript includes a Methods section, that is presented before the Results section. This part is too brief and overall unclear. In fact, it does not present all methods. Part of the analysis is presented throughout the Results section. As a result, the overall plan of the work is not clear, as well as its objectives. The study is articulated in several analyses, including mechanistic simulations of synthetic population systems with increasing level of complexity and the study of the covid-19 epidemic with data analysis techniques. The needs for combining all the different steps should be better explained since the beginning.

Later in the Results section, authors present the use of LOESS to fit the relationship between peak incidence and mobility. This analysis should be better presented, and the choice of the LOESS method should be better motivated.

In section “Effects of multi-seeding in the first pandemic wave” author present a correlation analysis between mobility flows and peak incidence. They write “We start our analyses by showing the connection between mobility from the region or province of origin to every destination and epidemic features such as the maximum local incidence or mortality in the area of destination”. The region “of origin” authors refers to is defined as the one that maximize the correlation. However, if I correctly interpreted this part, no validation is provided on the fact that the specific region is the actual origin of the country’s epidemic. To avoid any ambiguity, authors should use a different notation. At the same time, they could discuss their findings in light of available information on key epidemic events or first epidemic clusters detected in the countries. This is done with some extent, but it should be expanded and relevant citations should be provided. For instance, the first cluster identified in Germany was in Bavaria (https://www.thelancet.com/journals/laninf/article/PIIS1473-3099(20)30314-5/fulltext). Likely, similar information is available for other countries.

What is the role played by the population size of a destination region? This should impact the number of new cases at the peak, but also the timing of the peak. Would that be possible to provide a mechanistic understanding on its role and its interplay with mobility?

Figure 1: It seems to me that results with lower beta are quite interesting and would deserve to be presented in the main paper.

The sentence “Therefore, mobility yields a first-order effect on the local incidence peaks due to the multi-seeding, as we observed empirically, and synchronized and homogeneous lockdowns allow to observe this effect, potentially saving many lives.” is unclear.

The area considered in the analysis are either province or regions according to the country. However, authors use sometimes the term “province” to generically refer these areas (for all countries) thus creating confusion (e.g. caption of Figure 7)

**Have the authors made all data and (if applicable) computational code underlying the findings in their manuscript fully available?**

Reviewer #1: Yes

Reviewer #2: Yes

Reviewer #3: **No: **the section "Availability of data and materials" provides information about how to access the data. But the information regarding the code for the analysis is missing.

PLOS authors have the option to publish the peer review history of their article (what does this mean?). If published, this will include your full peer review and any attached files.

Reviewer #1: No

Reviewer #2: No

Reviewer #3: No
---

## [Editor Report · Decision Letter 1]

6 Aug 2021

Dear Dr Ramasco,

We are pleased to inform you that your manuscript 'Interplay between mobility, multi-seeding and lockdowns shapes COVID-19 local impact' has been provisionally accepted for publication in PLOS Computational Biology.

Best regards,

Alex Perkins

Associate Editor

PLOS Computational Biology

Tom Britton

Deputy Editor

PLOS Computational Biology

---

## [Editor Report · Acceptance letter]

20 Sep 2021

PCOMPBIOL-D-21-00481R1 

Interplay between mobility, multi-seeding and lockdowns shapes COVID-19 local impact

Dear Dr Ramasco,

I am pleased to inform you that your manuscript has been formally accepted for publication in PLOS Computational Biology. Your manuscript is now with our production department and you will be notified of the publication date in due course.

With kind regards,

Amy Kiss
